# Magnetization switching in ferromagnets by adsorbed chiral molecules without current or external magnetic field

Oren Ben Dor[1], Shira Yochelis[1], Anna Radko[1], Kiran Vankayala[2], Eyal Capua[2], Amir Capua[3], See-Hun Yang[3], Lech Tomasz Baczewski[4], Stuart Stephen Papworth Parkin[3,5], Ron Naaman[2] & Yossi Paltiel[1]

Ferromagnets are commonly magnetized by either external magnetic fields or spin polarized currents. The manipulation of magnetization by spin-current occurs through the spin-transfer-torque effect, which is applied, for example, in modern magnetoresistive random access memory. However, the current density required for the spin-transfer torque is of the order of $1 \times 10^6 \, A \cdot cm^{-2}$, or about $1 \times 10^{25}$ electrons $s^{-1} \, cm^{-2}$. This relatively high current density significantly affects the devices' structure and performance. Here we demonstrate magnetization switching of ferromagnetic thin layers that is induced solely by adsorption of chiral molecules. In this case, about $10^{13}$ electrons per $cm^2$ are sufficient to induce magnetization reversal. The direction of the magnetization depends on the handedness of the adsorbed chiral molecules. Local magnetization switching is achieved by adsorbing a chiral self-assembled molecular monolayer on a gold-coated ferromagnetic layer with perpendicular magnetic anisotropy. These results present a simple low-power magnetization mechanism when operating at ambient conditions.

[1] Applied Physics Department and the Center for Nano-Science and Nano-Technology, The Hebrew University of Jerusalem, Jerusalem 9190401, Israel. [2] Department of Chemical Physics, The Weizmann Institute of Science, Rehovot 7610001, Israel. [3] IBM Research Division, Almaden Research Center, 650 Harry Road, San Jose, California 95120, USA. [4] Magnetic Heterostructures Laboratory, Institute of Physics Polish Academy of Sciences, Al. Lotnikow 32/46, 02-668 Warszawa, Poland. [5] Max Planck Institute for Microstructure Physics, Halle (Saale) D-06120, Germany. Correspondence and requests for materials should be addressed to Y.P. (email: paltiel@mail.huji.ac.il).

Ferromagnets are magnetized permanently in domains that are either parallel or antiparallel to the easy axis directions. The magnetization direction can be manipulated using external magnetic fields or spin currents. Using spin currents, that is, electrons spin, could reduce scattering and enhance the devices operational frequency. The Spin currents are commonly employed to magnetize magnetic storage devices or for logical operations utilizing the spin-transfer torque (STT) effect[1,2]. In this effect the spin angular momentum of the electrons is transferred to the magnetic moments of the ferromagnetic (FM) film. Spin-based devices use the STT effect to reorient magnetization, such as in magnetoresistive random access memory (MRAM)[3]. Two main problems restrict the use of the STT effect in operating devices: First, the STT usually requires the use of complicated materials and systems, and second, the STT effect efficiency is low. Therefore, large electronic currents are required for inducing magnetization. Usually, the currents are of the order of $1 \times 10^6$ A cm$^{-2}$, or about $1 \times 10^{25}$ electrons s$^{-1}$ cm$^{-2}$ (refs 2,4). This relatively high current density significantly reduces the total device efficiency and enhances heating effects.

In previous works, it was found that when chiral molecules are self-assembled on gold, the interface has paramagnetic properties[5–8]. A similar process occurs in the Magnetism Induced by Proximity of Adsorbed Chiral molecules, or MIPAC effect. Owing to spin-selective electron transfer, the FM layer underneath the molecules becomes spin-polarized and hence magnetized. The magnetization reversal is achieved despite the fact that less than one electron is transferred per adsorbed molecule. The MIPAC effect originates from the well-known phenomenon in which the formation of a self-assembled monolayer (SAM) of molecules with a large dipole moment involves charge transfer that equalizes the electrochemical potential of the adsorbed layer and the sample surface[9,10]. The charge transfer induced by the SAM formation is responsible for the interesting room temperature (RT) magnetic properties observed when thiolated molecules are adsorbed onto gold[2,11,12].

In recent years, it has been established that electron transfer through chiral molecules is spin selective[13,14], namely, specific spin orientation is preferred in the transfer process. This effect is referred to as chiral-induced spin selectivity (CISS) where the spin is polarized either parallel or anti-parallel to the electrons' velocity according to the handedness of the molecules. Hence, as a result of the CISS effect, spin-polarized electron current is produced that can be used to magnetize ferromagnets. This concept was demonstrated in optical[15] and electrical[16] based devices by driving electrons through chiral layers.

In this work we combined the two properties, the SAM adsorption-induced charge transfer and the CISS effect, to demonstrate the ability to magnetize a FM layer by adsorption of a SAM made from chiral molecules. These results demonstrate magnetization switching of FM thin layers, which is induced solely by adsorption of chiral molecules (magnetism induced by proximity of adsorbed chiral molecules), where less than $10^{13}$ electrons per cm$^2$ are sufficient to induce the magnetization reversal on a gold-coated thin FM layer with perpendicular magnetic anisotropy. The locality of magnetization switching is achieved by adsorbing the chiral molecules as a self-assembled monolayer (SAM) in selective adsorption areas. The direction of the magnetization depends on the handedness of the adsorbed chiral molecules.

## Results

**Magnetic force microscopy probing.** The FM structure used was $Al_2O_3$/Pt/Au/Co (thickness-1.5–2.2 nm)/Au (thickness-5 nm), which is perpendicularly magnetized in the out-of-plane (OOP) direction. Different Co layer thicknesses were employed, resulting in different coercive fields (Supplementary Fig. 1) and domain sizes[5,17,18].

The SAMs were made with two enantiomers of the oligopeptide, which are based on α-helix polyalanine L and D (see Methods section). Both oligopeptides were adsorbed onto predetermined areas on the top gold layer, which caps the under-layer and prevents its oxidation. The L and D enantiomers are referred to as AHPA-L and AHPA-D, respectively, throughout this paper.

The electroresist-coated sample was patterned using electron-beam (e-beam) lithography. The molecules were adsorbed only in the designated areas, after removing the electroresist (Fig. 1). Atomic force microscopy (AFM) and X-ray photospectroscopy (XPS) were used to verify the adsorption of the SAM. Different selective adsorption area sizes were tested ranging from $50 \times 50$ nm$^2$ to $1 \times 1$ µm$^2$ (Fig. 1b).

The local magnetization was measured using magnetic force microscopy (MFM), a Hall probe and a superconducting quantum interference device with vibrating sample magnetometer (SQUID-VSM). During all MFM measurements the tip scanned at an elevated height, $\sim 25$–30 nm above the surface, to prevent the topography from influencing the measurement.

The magnetic-induced phase shift in the MFM was measured on the same selective adsorbed areas as the topography ones, as shown in Fig. 2. The slight differences in height between AHPA-L and AHPA-D topography images are attributed to different synthesized purity levels and consequently, to differences in self-assembled monolayer quality. Although the topography appears similar for the two enantiomers (Fig. 2a,b), the measured magnetization is different and correlates with the handedness of the chiral molecules, namely, OOP magnetization that is parallel to the molecules' major axis for AHPA-L (Fig. 2c) and anti-parallel to the molecules' major axis for AHPA-D (Fig. 2d). The results were reproducible for many samples and various thicknesses of the cobalt FM layer, and were observed even when the molecules were adsorbed by drop-casting (Methods secion and Supplementary Fig. 2). When the magnetization was recorded over time, we found that it diminished after about nine days. This was true only for samples that were not fully covered with the chiral molecules' monolayer.

By reducing the dimensions of the selective adsorption area, it was possible to demonstrate a switching of the magnetization on the nanometer scale. The magnetization switching at RT could be observed down to a single domain size. The size of the domains is related to the thickness of the FM layer; therefore, in our samples the minimum size was $\sim 250$ nm (Fig. 3a). The inset of Fig. 3a shows the average domain size without the adsorbed chiral molecules. Since the FM layers are thin, we expect Néel walls and not Bloch walls between magnetic domains, favouring efficient domain wall motion in the plane[19]. This property may be useful for moving magnetically induced matrices, thus creating surface memory logic attributes.

When we attempted to probe selective adsorption areas smaller than 250 nm, that is, smaller than the size of a domain in our samples, we managed to see the surface topography but not the MFM magnetic phase image. We attribute this outcome to the inability to reorient magnetization in an area smaller than a single domain size. Furthermore, the MFM magnetic phase measurement of selective adsorption areas patterned on a sample with a very large coercive field on the order of 3,000 Oe (an order of magnitude larger than the proximity-induced coercivity) showed very little response (Supplementary Fig. 3). These results further support the existence of the FM-chiral SAM coupling.

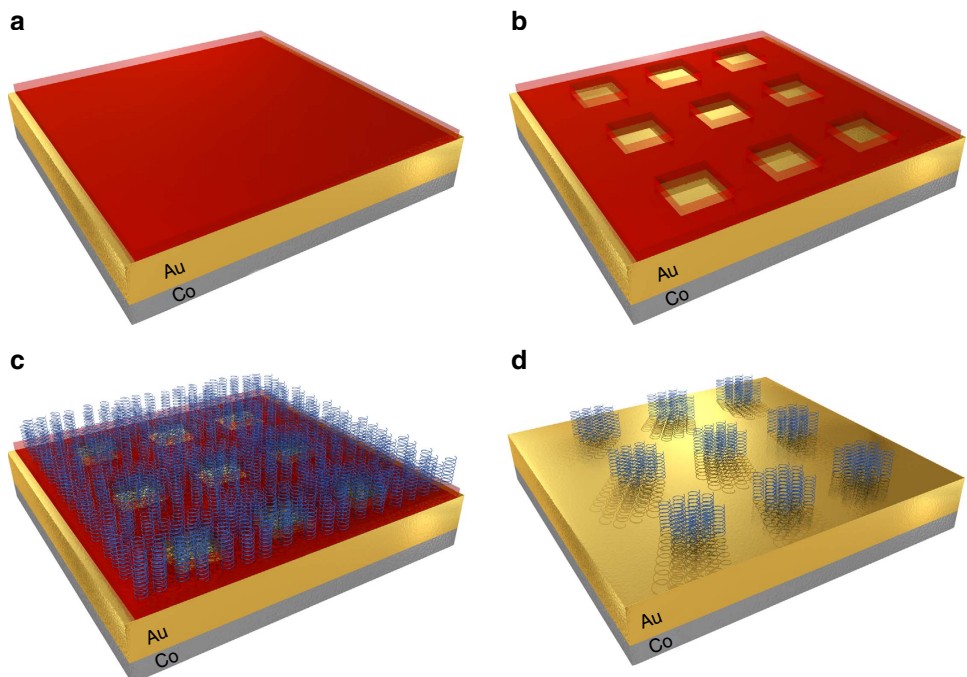

**Figure 1 | Selective adsorption.** (**a**) Poly(methyl methacrylate) (PMMA) was spin coated on the top of the sample (red) with the top two layers made of Cobalt (grey) and Gold (yellow). (**b**) Selective areas were patterned using e-beam lithography. (**c**) The sample was immersed in an ethanolic solution containing the organic molecules (AHPA-L or AHPA-D) (blue). (**d**) The molecular residue and remaining PMMA were removed by rinsing in ethanol and acetone several times under inert conditions. Selective adsorption areas ranged from $50 \times 50$ nm$^2$ to $1 \times 1$ μm$^2$ squares. The molecules were adsorbed at a $40°$ angle relative to the surface.

**Anomalous Hall Effect**. In order to test a variety of FM thin samples, and because the molecular beam epitaxy (MBE)-grown Co samples are difficult to pattern, additional Anomalous Hall Effect (AHE) measurements[20,21] were conducted on 7 nm Ni FM e-beam evaporated film, capped with a thin 2 nm gold over-layer (see Methods section). Here too, a similar molecular proximity-induced magnetization behaviour was obtained using this measurement technique with a different magnetic film. It was previously shown that an evaporated 7 nm Ni film on a Si substrate has an OOP easy axis[22]. The thin FM Ni layer acts as the conductive channel in which the AHE occurs. As in the MFM measurements, the adsorption of molecules resulted in magnetization switching of the FM layer. Initially, a calibration measurement of the Hall coefficient and the Hall probe asymmetry was done before molecular adsorption was performed[15]. Later, the molecules were drop-casted onto the device and dried under inert conditions, without any additional changes made in the electrical setup. Hall measurements were performed after molecules were chemisorbed onto the gold over-layer that covers the conductive Ni channel, resulting in a constant induced magnetic field. Hence, the unchanged nature of the Hall resistance during the time scale shown in Fig. 4c is the manifestation of that fixed induced magnetic field. The resistance $R_{xy} = \frac{V_y}{I_x}$ was measured, where $V_y$ is the potential measured between the Hall electrodes and $I_x$ is the source-drain current.

The results of the AHE measurements are presented in Fig. 4, accompanied by a scheme describing the experimental setup. Since the AHPA-L and AHPA-D molecules induced magnetization in opposite directions, the Hall voltage, $V_y$, in both cases has an opposite sign, resulting in a change of sign of $R_{xy}$. The asymmetry between the values measured for the two enantiomers may result from asymmetry in the current flow through the Hall channel due to differences arising from the fabrication process. Several devices were tested, all of which showed opposite $\Delta R_{xy}$ signs for the two different enantiomers.

**Magnetic moment analysis**. The MIPAC effect was also verified using a SQUID-VSM. In these studies, the FM layer was 10 Al$_2$O$_3$|2 TaN|1.5 Pt|0.2 Au|0.3 Co|0.7 Ni|0.15 Co|1 Au (thicknesses in nanometers). Here the magnetization of the sample was first saturated and then the molecules were deposited. The sample size was $4 \times 4$ mm$^2$ and had a saturation magnetic moment of $\pm 1.5 \times 10^{-5}$ e.m.u., where the sign refers to the direction of the magnetic moment, namely, the plus sign denotes a magnetic moment pointing away from the surface, whereas the negative sign refers to the moment pointing towards the surface. We saturated the magnetization in each sample using an external magnetic field, and then adsorbed the chiral molecules with the opposite sign, which should reverse the magnetization. This was done for both directions of magnetic field. Samples were measured with and without molecules. The molecular-induced magnetization was compared to the saturated magnetization of the sample. We found that the adsorption of the chiral molecules changed the magnetic moment of the sample by about 35% relative to the saturation magnetization, with a positive sign for AHPA-L and a negative one for AHPA-D. Taking into account the molecular density of $10^{13}$ molecules per cm$^2$, sample size and the 35% molecular-induced change in magnetic moment $0.525 \times 10^{-5}$ emu, we can deduce that the magnetic moment per molecule is $3.28 \times 10^{-18}$ e.m.u. Since the MFM measurements show no domain formation under the chiral adsorbed layer, we relate the partial magnetization reversal to the rotation of the magnetic easy axis due to the molecular adsorption angle, as was measured in other materials[23,24]. The angle of the chiral molecules was measured to be $40°$ relative to the surface normal. It is important to realize that the orientation of the molecules is defined by this angle and therefore the average direction of magnetization will be perpendicular to the surface normal but with a lower value. Figure 4c inset shows $M(H)$ magnetization loop for the sample, comparing between the magnetization loop after AHPA-L adsorption and bare samples.

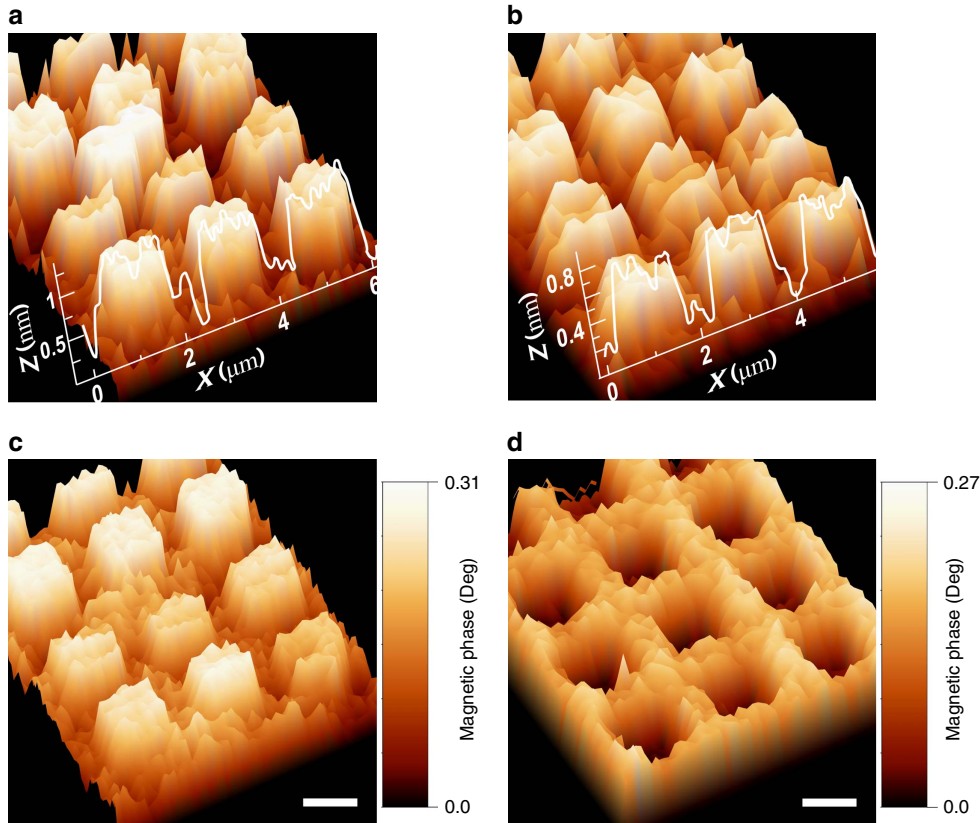

**Figure 2 | Topography and magnetic phase images.** AFM topography images of the SAMs of AHPA-L (**a**) and AHPA-D (**b**) adsorbed onto ferromagnetic Co thin layers, including $z$ (nm) and $x$ (μm) scale bars, coated with a 5 nm gold over-layer and their corresponding magnetic force microscopy molecular-induced magnetic phase images and colour bars of AHPA-L (**c**) and AHPA-D (**d**) magnetization orientation, displaying the same area as seen in the topography images (scale bar is set to 1 μm).

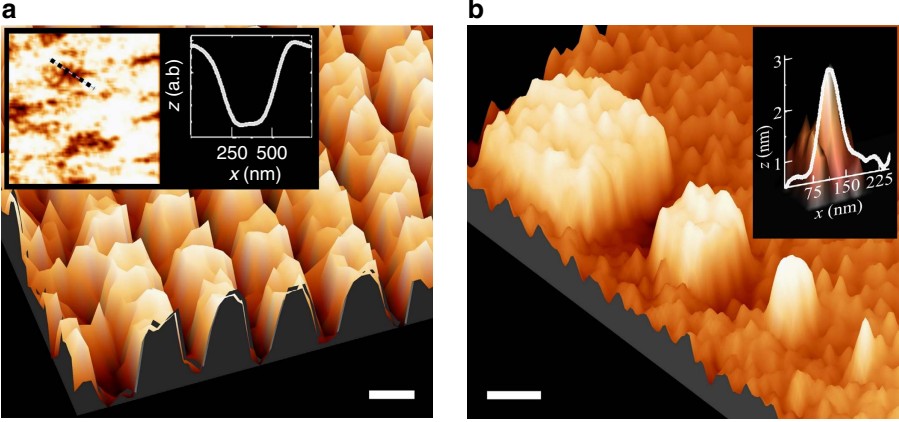

**Figure 3 | Device topography and molecular-induced magnetization.** (**a**) Magnetic force microscopy magnetic phase image of a molecular-induced magnetization direction by AHPA-L chiral molecules in a matrix of $250 \times 250 \, nm^2$ selective adsorption areas (scale bar is set to 300 nm). The inset shows magnetic domain sizes before adsorbing the molecules (left) with a cross section view (right) along the dashed line. (**b**) Atomic force microscopy topography image of $1 \times 1 \, \mu m^2$, $500 \times 500$, $250 \times 250$, $100 \times 100$ and $50 \times 50 \, nm^2$ (the latter is shown in the inset) chiral self-assembled monolayer selective adsorption areas (scale bar is set to 500 nm).

The AHPA-L adsorption shifted the magnetization loop by 10–15 Oe, which is around 50% of the full magnetization value of the sample.

## Discussion

The experimental results show that adsorbed chiral molecules induce magnetization switching in the FM samples at RT. We ascribe the observed results to a proximity effect where the

wavefunction of the electrons in the FM is coupled to the SAM through the Au layer (Fig. 5). Since only the wavefunction with the preferred spin orientation can penetrate into the SAM, the density of states in the FM is not the same for the two spin orientations and as a result, magnetization in a specific direction is observed. This model explains the dependence of the switching direction upon the adsorption of the two enantiomers. Similarly to the CISS effect, here too the delocalized wavefunction is

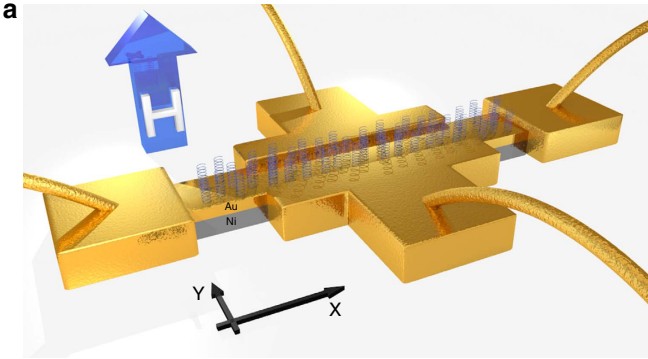

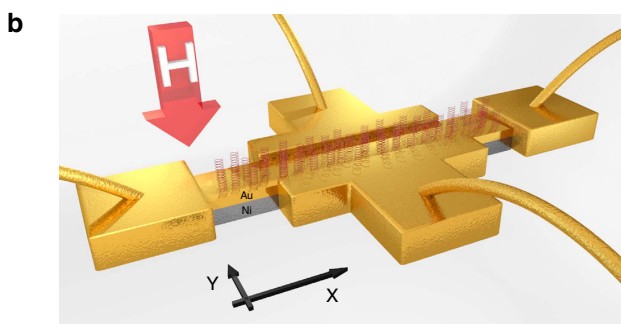

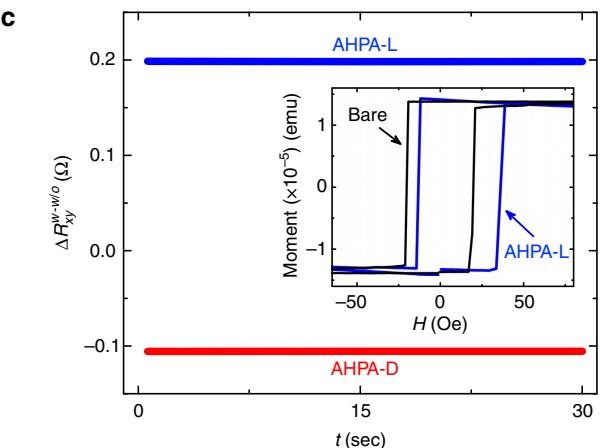

**Figure 4 | Anomalous Hall effect device configuration scheme and results.** Current $I_x = 1\,mA$ is driven along the x-axis, voltage, $V_y$, is measured along the y-axis at room temperature. (**a,b**) AHPA-L (AHPA-D) chiral molecules adsorbed onto the conducting channel induce upwards (downwards) magnetization **H** as denoted by a blue (red) arrow, respectively. (**c**) The difference between $R_{xy}$ for a bare sample and $R_{xy}$ after AHPA-L (blue) and AHPA-D (red) adsorption. The unchanged Hall resistance seen during the time of measurement is the manifestation of a constant induced magnetic field caused by the molecular adsorption. Different devices display different asymmetries although all have a consistent $\Delta R_{xy}$ opposite sign for both molecule types. (**c**, inset) Superconducting quantum interference device with vibrating sample magnetometer $M(H)$ data for the AHPA-L adsorbed and bare configurations of the thin multilayer sample. The AHPA-L adsorption shifted the magnetization loop by 10–15 Oe, which is around 50% of the full magnetization value of the thin multilayer sample (not the anomalous Hall effect sample).

influenced by the molecules' chirality. The delocalization of sample surface states, upon adsorption of a top layer, is well documented in the literature[17,18]. Weak adsorption-induced paramagnetic behaviour was observed when thiolated molecules

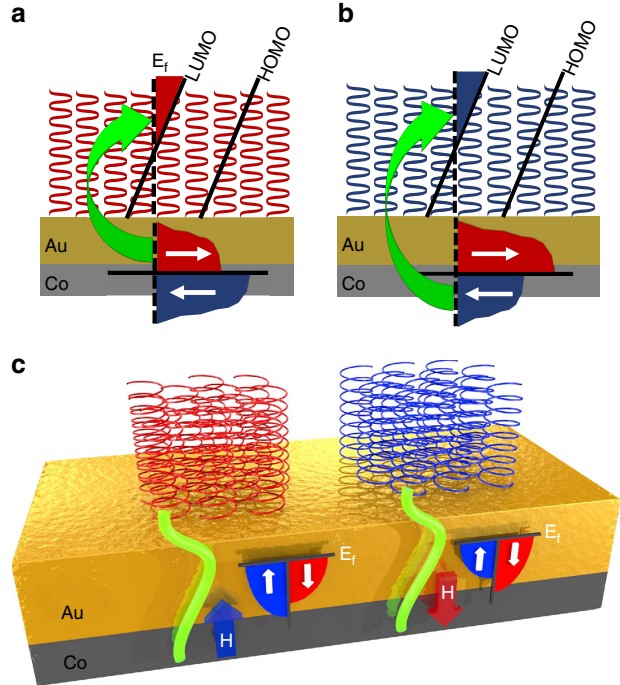

**Figure 5 | Experimental concept.** In the semi-classical approach the adsorption induces transfer of electrons with a specific spin from the FM layer to the SAM (denoted by the green arrow). Owing to the SAM dipoles, the LUMO is partially below the Fermi level (dashed line). Electrons are transferred to the SAM to fill the now empty states below the Fermi energy. The transferred electrons have a specific spin because of the chirality of the AHPA-D (**a**) or AHPA-L (**b**) molecules. Consequently, splitting in the spin states' density in the FM is created, hence directional magnetization. (**c**) In the suggested quantum mechanical approach the electronic wave function (green) delocalization induces imbalances in spin-up (blue) and spin-down (red) populations, indicated as anisotropic magnetization **H** pointing upwards or downwards, respectively.

were adsorbed onto gold[5–8], and the effect of wavefunction penetration from the surface to the adsorbed layer was calculated for various organic-inorganic interfaces[25]. From a semi-classical perspective in can be understood that when chiral molecules are adsorbed onto a gold layer covering a Co thin film, the SAM dipoles lower the lowest unoccupied molecular orbital state (LUMO) to be partially below the Fermi level. As a result, electrons are transferred to the SAM and fill the empty states. The transferred electrons are spin-polarized due to the molecules' chirality. Consequently, stimulated spin-splitting takes place, which dictates the directionality. It was found that up to one electron per molecule can be transferred[26]. Therefore, with the SAMs used in the current study, this results in $10^{13}$ electrons$\cdot$cm$^{-2}$ that are being transferred. Note that the electrons that transfer to the LUMO state lose coherence much faster than the FM layer electrons.

The results obtained from the SQUID-SVM measurements indicate that the FM magnetization easy axis rotates due to the MIPAC effect. This may explain the reduction in magnetization over time if the film is only partly covered by molecules. This partial coverage leads to greater tilting of the molecules and as a result, to weaker FM molecular coupling. With time, the neighbouring FM domains orient the magnetization according to the original orientation. This is attributed to the exchanged interaction that causes the domains outside the selective adsorption areas to gradually reorient the molecular-induced magnetization back to an average state. Longer-lasting

magnetization could probably be achieved by changing the ferromagnetic materials and linkers' angle. The fact that MIPAC affects a very thin layer can be utilized for the specific magnetization of one thin layer in a multilayer structure. Selective etching and adsorption of local chiral molecules can be used to write three-dimensional matrices (Supplementary Fig. 4), which can be moved by domain wall motion.

To summarize, MIPAC presents a way to achieve simple and local magnetization on ferromagnetic material. MIPAC is a surface effect in which OOP magnetization is induced in FM thin films by the adsorption of chiral molecules. The magnetization is realized without the use of an external magnetic field or electrical current. The direction of magnetization depends on the chirality of the adsorbed molecules. The magnetization can be induced on preselected small areas by controlling the adsorption of the chiral molecules. In principle, the magnetization can be localized to an area as small as a single domain, thereby overcoming the tendency of FM materials to become super-paramagnetic on shrinking their size to the dimensions of a single domain. This concept can be used to achieve simple surface spintronic logic devices.

## Methods

**MFM probing of selective adsorption areas.** Thin film samples were grown by the molecular beam epitaxy (MBE) method, in the configuration $Al_2O_3$ (0001)/Pt (5 nm)/Au (20 nm)/Co (1.5–2.2 nm)/Au (5 nm). A Pt buffer layer was deposited at 650 °C to ensure an atomically flat buffer layer surface—sharp streaks were visible in the RHEED image. Crystallographic orientation of the grown layers was as follows: Pt(1110), Au(111), Co(0001) and Au(111). Pt was deposited using an electron gun; Co and Au—from effusion cells. Samples differed in their Co thickness, ranging from 1.5 nm (used most often) to 2.2 nm, thus resulting in different coercive field $H_c$ values, but all had rectangular hysteresis loops for a magnetic field direction perpendicular to the plane, which is characteristic of the OOP easy axis. Hysteresis loops were measured by a polar magneto-optic Kerr effect (P-MOKE) for each sample (Supplementary Fig. 1).

Sample cleaning was performed with N-Methyl-2-pyrrolidone (NMP) at 80 °C for 10 min, followed by washing in isopropanol and water. Owing to FM layer sensitivity to exerted forces, no aggressive cleaning was done (for example, no plasma asher or ozonator). Alignment marks were evaporated in order to locate the adsorption areas, during which samples were spin coated with conventional lift-off resist (LOR) followed by a hard bake for 5 min at 200 °C. After cool down, the samples were then spin-coated with AZ1505 photoresist followed by an additional bake for 90 s at 110 °C. Alignment marks were patterned by Laser writer Microtech LW405B and later developed by AZ726 developer. Marks were produced by evaporation of 15 nm Cr and 150 nm Au at −5 °C, followed by a conventional liftoff procedure of 10 min in NMP at 80 °C, washing in isopropanol and rinsing in water. To pattern selective adsorption areas, PMMA electroresist was spin-coated onto the sample surface followed by a 2 min hard bake at 180 °C, resulting in a 44nm coating layer (Fig. 1a). Test samples that underwent this entire procedure exhibited unchanged magnetic behaviour, suggesting that all samples used were magnetically intact. An e-beam was used to pattern the selective areas of different sizes ranging from $1 \times 1$ μm$^2$ to $50 \times 50$ nm$^2$ (Fig. 1b). Here we used SAMs of opposite chirality: AHPA-L and its mirror image AHPA-D, which are based on α-helix L and D amino acids (H-CAAAAKAAAAKAAAAKAAAAKAAAAAKAA AAKAAAAK-OH), where C, A, and K represent cysteine, alanine and lysine, respectively. Adsorption was performed under inert conditions. Initially, samples were immersed in absolute ethanol for 20 min, dried in nitrogen and later placed into a solution consisting of 1 mM AHPA-L (or D) in ethanol and left overnight (Fig. 1c). Following the SAM adsorption, the samples were immediately washed in absolute ethanol to remove the un-bonded molecular residue, followed by washing in absolute acetone to remove the remaining PMMA layer, and finally bathed in ethanol again (Fig. 1d). This cleaning procedure was repeated several times (for both AHPA-L and AHPA-D). Although some resist residue still remained, and appeared corrugated in the force microscopy images, the AFM images clearly show the adsorbed SAMs inside the selective e-beam-patterned areas (Fig. 2a,b).

During the MFM measurements, simultaneous topography and magnetic phase images were obtained. The samples were investigated by Atomic Force Microscopy (D3100, Nanoscope V controller, Bruker, Santa Barbara, CA, USA). The AFM non-contact mode and tapping mode were used to determine the magnetic domains simultaneously with the surface topography of the samples. The images, ranging from $1 \times 1$ to $10 \times 10$ μm$^2$, were obtained in air by using Si probes with Co-Cr coating (MESP, k ∼ 5 N/m, F ∼ 75 kHz, Bruker). Interleave scanning was applied to obtain the magnetic images. For scanning, we raised the tip height to ∼ 25–30 nm from the surface to prevent the topographic and magnetic forces from interacting. The phase-detection mode was used to collect the magnetic signal,

where the signal of the phase shift was around 1.2°. Without a magnetic field, the phase shift was zero, corresponding to the maximum of the resonance peak.

**Anomalous Hall Effect measurements.** The AHE device was constructed from a $4 \times 40$ μm$^2$ Hall channel made from 7 nm Ni capped with 2 nm Au on top of a thermally induced SiO$_2$ surface. The Au over-layer prevents oxidation of the FM sub-layer. A current of $I_x = 1$ mA was driven along the channel (only in the Ni layer) while Voltage, $V_y$, was measured perpendicular to the current direction (Fig. 4a,b). The AHE device was placed on a chip and its contact pads were bonded to the chip using gold ball bonders. During the AHE measurements, the device remained attached to its carrying chip, where 4 μl of the 1 mM molecules in ethanolic solution were drop-casted (Supplementary Fig. 2) onto the top of the device and later dried in an inert environment. Drop-casting is used to prevent changes in the assembly of the bonders connecting the sample to the chip, thus ensuring that any change in the Hall Voltage readout is only due to the added molecules.

**Data availability.** The data that support the findings of this study are available from the corresponding author on request.

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

## Acknowledgements

Y.P. acknowledges support from the Volkswagen Foundation (VW 88 367), from the Israel Science Foundation (ISF grant no. 1248/10), and the MOS Israel. R.N. acknowledges support in part from the European Research Council under the European Union's Seventh Framework Program (FP7/2007–2013) / ERC grant agreement no (338720), from the MOS Israel and from the VW Foundation (VW 88 367). O.B.D. would also like to acknowledge the Israeli Ministry of Science, Technology and Space grant 0399174. Y.P. and O.B.D. thank I. Eisenberg for his helpful and professional graphical contribution to this paper.

## Author contributions

The manuscript was written with contributions from all authors. All authors have approved the final version of the manuscript and participated in the discussions related to the work. L.T.B. and A.C. were responsible for growth and characterization of the thin film samples using the M.B.E. method. O.B.D. and K.V. fabricated and characterized the devices and performed the experimental work. A.R. performed the M.F.M. measurements. S.Y. and E.C. were in charge of the chemical aspects of the paper. Y.P. and R.N. supervised the work. O.B.D., S.Y., R.N. and Y.P. developed the narrative and major concepts presented in the paper.

## Additional information

**Competing financial interests:** The authors declare no competing financial interests.

**Publisher's note**: 

