## [Peer Review File · Nature Communications]

Reviewers' comments:

Reviewer #1 (Remarks to the Author):

This manuscript demonstrates magnetization switching of perpendicularly magnetized metal films induced by surface adsorption of chiral molecules. The magnetization switching was found to depend on the chirality of the surface molecules, which was evidenced by a combination of magnetic (MFM and SQUID magnetometry) and electrical (Hall effect) measurements.

Overall the experiments were well motivated and nicely designed. The results, as reported in the manuscript, provided rather compelling evidence for the effect of chiral molecule SAM induced magnetic switching. To my knowledge, the primary finding of the work would represent a first demonstration of magnetization reversal without an applied magnetic field or active spin transfer. The effect is not only interesting at the fundamental level, but also could potentially be the basis for new device applications. Before I can recommend publication of the manuscript, however, I would like to see clarification of some key experimental details:

1) For the AFM/MFM measurements in Fig 2, what are the scales for the topography images (a and b)? The images for AHPA-L and AHPA-D appear quite different. Line scans with height scales should be provided for clarification. Are the AFM and MFM data obtained from the same sample and areas?

2) How were the Hall measurements performed? The manuscript shows data of change of 'Hall resistance' upon drop-casting of AHPA-L and AHPA-D solutions as a function of TIME. Is the data in Fig 3c simply a change of V_y upon molecule deposition without an applied magnetic field? It is well known that even in lithographically patterned Hall bar geometry, V_y always contains a longitudinal component in addition to the transverse Hall component. To obtain the Hall component, one needs to extract the anti-symmetric component of the V_y vs B curve. Without a complete R_{xy} vs B curve, one cannot definitively attribute the 'change of Hall resistance' to variation of the anomalous Hall component; this change could originate from variation of the longitudinal component or the ordinary Hall effect due to a change in the carrier density upon molecular adsorption.

3) Similarly, without showing any $M(H)$ data, the manuscript claims changes of the magnetization by approximately $\pm 35\%$ for AHPA-L and AHPA-D adsorption respectively. Moreover, the authors attributed the magnetization changes to the rotation of the magnetic easy axis due to the chiral molecule adsorption. The $M(H)$ data need to be presented and analyzed in the manuscript in order to back up these claims. Also, the numbers quoted for 'magnetization' (e.g., line 143) are in fact for 'magnetic moment'. It would be more informative to present the actual magnetization. Especially, what is the induced moment change per molecule?

In summary, in my judgment, the central result reported in the manuscript is very interesting and potentially important, however, it needs to be established with more rigor. Some of the key experimental data should be presented in the manuscript and discussed in more depth. As is, I do not recommend publication of the manuscript.

Reviewer #2 (Remarks to the Author):

The paper reports that magnetization switching can occur when chiral molecules are adsorbed on a ferromagnetic surface. This effect is quite interesting and follows the effects already reported by Naaman and co-workers about the so called chiral-induced-spin selectivity. I recommend acceptance of the paper. Still, I have some questions that should be clarified before an eventual publication of these results: 1) The authors check this magnetization reversal using various techniques (MFM, anomalous Hall effect, SQUID-VSM), but in each case the FM structure is different and also the nature of the ferromagnetic layer (Co or Ni). The authors should explain the reasons of these choices. 2) In all cases the layer in direct contact with the SAM molecules is a gold layer. Why is this layer needed? What happens if this layer is removed? As it is now, it seems that the combination gold-chiral molecules is the required ingredient to observe such an effect.

Reviewer #3 (Remarks to the Author):

Manuscript describes experimental manifestation of a new effect of magnetism induced by proximity of adsorbed chiral molecules (MIPAC). Authors claim that charge transfer between adsorbed molecules and ferromagnetic substrate together with spin selectivity induced by molecules' chirality are the origins of the demonstrated magnetization switching in the substrate.

The finding is very exciting and clearly new: the observed effect does not require either magnetic fields or spin polarized currents to induce the switching. Moreover, authors estimate that transfer of only 10^{13} electrons per cm^2 is required to induce switching contrary to spin current densities of 10^{25} electrons per $\text{cm}^2 \text{ s}$ in the usual switching setup. In my opinion, the work clearly deserves to be published in the Nature Communications.

There is a minor issue the authors may want to clarify before the publication. There are two statements in the manuscript. First (on p.4) is about finite lifetime of the induced magnetization (for samples not fully covered with the chiral molecules it diminishes in nine days). Second (on p.5) is about inducing magnetization in small areas (down to a single domain size) with a prospect of utilizing this local magnetization to build surface memory logic. It is a bit confusing if the magnetization induced locally will be stable. And if yes, what is the difference from the situation mentioned in the first statement? Is it due to domain formation?

We thank all 3 referees and the editor for their very positive reports, and for their useful and important comments. We were happy to notice that all 3 referees are in favor of publishing the paper after revision.

It what follows we respond point-by-point to the referees' comments. Our changes in the manuscript are marked in yellow.

Reviewer #1

a) Before I can recommend publication of the manuscript, however, I would like to see clarification of some key experimental details:

1) For the AFM/MFM measurements in Fig 2, what are the scales for the topography images (a and b)? The images for AHPA-L and AHPA-D appear quite different. Line scans with height scales should be provided for clarification. Are the AFM and MFM data obtained from the same sample and areas?"

Author's response:

The referee is right and indeed the topography scales were missing. Height scales were therefore added to Figures 2a,b with a brief explanation in **line 116**.

Indeed, there are visible differences between AHPA-L and AHPA-D AFM topography images. It is well known [See for example: Carmeli et. al. Angew. Chem. 41,761-764 (2002).] and well documented that it is easier to obtain pure AHPA-L since they are based on natural pure L enantiomers while the D enantiomer is synthesized as a racemic mixture containing both L and D

enantiomers and only then separated. Hence in the case of D the probability that the oligomer will contain the “wrong” chirality is not negligible and therefore the packing of the oligomer is less ideal, as seen in the AFM topographical image. This is now also explained in the manuscript (lines 82-87). The referee is also right that we did not indicate directly that the AFM and corresponding MFM images display the exact same probing area. This was added in yellow marking in lines 81-82, 118-119, Supplementary: 3-4, 14-15.

b) 2) How were the Hall measurements performed? The manuscript shows data of change of ‘Hall resistance’ upon drop-casting of AHPA-L and AHPA-D solutions as a function of TIME. Is the data in Fig 3c simply a change of V_y upon molecule deposition without an applied magnetic field? It is well known that even in lithographically patterned Hall bar geometry, V_y always contains a longitudinal component in addition to the transverse Hall component. To obtain the Hall component, one needs to extract the anti-symmetric component of the V_y vs B curve. Without a complete R_{xy} vs B curve, one cannot definitively attribute the ‘change of Hall resistance’ to variation of the anomalous Hall component; this change could originate from variation of the longitudinal component or the ordinary Hall effect due to a change in the carrier density upon molecular adsorption."

Author's response:

We agree that indeed the Hall measurements procedure was not described with enough details. A clarification regarding the Hall measurement method was added in lines 104-111, 139-141. Initially the device was calibrated without the molecules (differential Hall measurement method is further explained in ref. 15: Ben Dor, O., Morali, N., Yochelis, S., Baczewski, L. T. & Paltiel, Y. Local light-induced magnetization using nanodots and chiral molecules. *Nano Lett.* 14, 6042–6049 (2014)), only then molecules were drop casted.

After the molecules were chemisorbed onto the gold layer and the magnetization induced by molecular proximity effect was in equilibrium, i.e. inducing a constant magnetic field on top of the Hall channel, measurements were made comparing the Hall voltage on the same device with and without the molecules.

The final result, as shown in Figure 3c, is the subtraction of the initial molecules-free signal from

the signal with the molecules. The time scale in Figure 3c indicates that the molecular induced magnetic field remained constant and stable throughout the measurements.

- c) 3) *Similarly, without showing any $M(H)$ data, the manuscript claims changes of the magnetization by approximately +35% for AHPA-L and AHPA-D adsorption respectively. Moreover, the authors attributed the magnetization changes to the rotation of the magnetic easy axis due to the chiral molecule adsorption. The $M(H)$ data need to be presented and analyzed in the manuscript in order to back up these claims. Also, the numbers quoted for 'magnetization' (e.g., line 143) are in fact for 'magnetic moment'. It would be more informative to present the actual magnetization. Especially, what is the induced moment change per molecule?"*

Author's response:

The $M(H)$ data was added as inset of Figure 3c and **lines 142-145**. Also, the term 'magnetization' was changed to 'magnetic moment'. The information regarding the induced magnetic moment per molecule was added on **lines 183-186**.

- d) *"In summary, in my judgment, the central result reported in the manuscript is very interesting and potentially important, however, it needs to be established with more rigor. Some of the key experimental data should be presented in the manuscript and discussed in more depth. As is, I do not recommend publication of the manuscript."*

Author's response:

We are happy to see that the referee found the manuscript very interesting and potentially important. We added all the requested corrections the manuscript.

Reviewer #2

- a) *"The paper reports that magnetization switching can occur when chiral molecules are adsorbed on a ferromagnetic surface. This effect is quite interesting and follows the effects already reported by Naaman and co-workers about the so called chiral-induced-spin selectivity. I recommend acceptance of the paper."*

Author's response:

We thank the referee for a very positive review.

- b) *The authors check this magnetization reversal using various techniques (MFM, anomalous Hall effect, SQUID-VSM), but in each case the FM structure is different and also the nature of the ferromagnetic layer (Co or Ni). The authors should explain the reasons of these choices."*

Author's response:

This is indeed an important comment which was not explained properly in the manuscript. There are two reasons for using different FM materials: 1) We wanted to show that the effect is general and similar for many different FM structures. 2) Technical issues prevent the use of all samples in all three types of magnetic measurements. For example, the Co based thin samples were grown by MBE method and therefore are more difficult to pattern without damaging the sample. This is the reason why they were not used in the AHE measurements. The Ni used in the AHE samples was evaporated by conventional e-beam deposition and consequently they were much easier to pattern and measure in that configuration. A remark discussing these considerations was added (lines 96-100).

- c) *2) In all cases the layer in direct contact with the SAM molecules is a gold layer. Why is this layer needed? What happens if this layer is removed? As it is now, it seems that the combination gold-chiral molecules is the required ingredient to observe such an effect."*

Author's response:

The gold over-layer has two important roles, it acts as a protective layer against oxidation of the

FM sub-layer. It enables the formation of the chemically adsorbed self-assembled layer by binding the thiol groups at the end of the molecule with gold. Adsorption of thiols on gold is well documented and characterized and therefore we chose it for the present study. An explanation was added in **lines 62-63, 337-338**.

Reviewer #3

a) Manuscript describes experimental manifestation of a new effect of magnetism induced by proximity of adsorbed chiral molecules (MIPAC). Authors claim that charge transfer between adsorbed molecules and ferromagnetic substrate together with spin selectivity induced by molecules' chirality are the origins of the demonstrated magnetization switching in the substrate.

The finding is very exciting and clearly new: the observed effect does not require either magnetic fields or spin polarized currents to induce the switching. Moreover, authors estimate that transfer of only 10^{13} electrons per cm^2 is required to induce switching contrary to spin current densities of 10^{25} electrons per cm^2 s in the usual switching setup. In my opinion, the work clearly deserves to be published in the Nature Communications."

Author's response:

We wish to thank the referee for the warm and encouraging words.

a) There is a minor issue the authors may want to clarify before the publication. There are two statements in the manuscript. First (on p.4) is about finite lifetime of the induced magnetization (for samples not fully covered with the chiral molecules it diminishes in nine days)."

Author's response:

The referee raises an important issue which we yet not fully understand. We assume that the finite lifetime observed is due to the competition between the exchange interaction in the FM and the effect of the adsorbed molecules. Film reconstruction may be responsible for domains

boundary motion that results in losing the magnetic dipole orientation induced by the adsorbed layer. This notion is strengthened by our observation that the “magnetic decay” does not take place if all the sample is covered by the adsorbed molecules. Our suggested explanation is presented in **lines 222-228** in the manuscript.

b) Second (on p.5) is about inducing magnetization in small areas (down to a single domain size) with a prospect of utilizing this local magnetization to build surface memory logic. It is a bit confusing if the magnetization induced locally will be stable. And if yes, what is the difference from the situation mentioned in the first statement? Is it due to domain formation?"

Author's response:

We believe that the key to the problem raised by the reviewer is the structure of the sample and the quality of the adsorption, both were not optimized in the present study. Further investigation is required with other samples and different linkers that may yield different domain size, increase the molecule-FM layer interaction and therefore may result in a stable effect that does not decay with time or decays very slowly. We added this explanation in **lines 228-230**.

REVIEWERS' COMMENTS:

Reviewer #1 (Remarks to the Author):

I am satisfied with the authors' technical responses to my review. I recommend publication of the manuscript. There are still many grammatical errors and other language issues in the manuscript, I suggest a careful editing before publication.

Reviewer #3 (Remarks to the Author):

Authors answered all the questions. The manuscript can be recommended for publication.

We thank all 3 referees and the editor for their very positive reports, and for their useful and important comments. We were happy to notice that all 3 referees are in favor of publishing the paper after revision.

It what follows we respond point-by-point to the referees' comments. Our changes in the manuscript are marked in yellow.

Reviewer #1

a) This manuscript demonstrates magnetization switching of perpendicularly magnetized metal films induced by surface adsorption of chiral molecules. The magnetization switching was found to depend on the chirality of the surface molecules, which was evidenced by a combination of magnetic (MFM and SQUID magnetometry) and electrical (Hall effect) measurements.

Overall the experiments were well motivated and nicely designed. The results, as reported in the manuscript, provided rather compelling evidence for the effect of chiral molecule SAM induced magnetic switching. To my knowledge, the primary finding of the work would represent a first demonstration of magnetization reversal without an applied magnetic field or active spin transfer. The effect is not only interesting at the fundamental level, but also could potentially be the basis for new device applications."

Author's response:

We thank the referee for the warm words. We agree with the referee about the possible implications and applications of this current work.

b) Before I can recommend publication of the manuscript, however, I would like to see clarification of some key experimental details:

1) For the AFM/MFM measurements in Fig 2, what are the scales for the topography images (a and b)? The images for AHPA-L and AHPA-D appear quite different. Line scans with height scales should be provided for clarification. Are the AFM and MFM data obtained from the same sample and areas?"

Author's response:

We thank the referee for the remark and indeed the topography scales were missing. Height scales were therefore added to Figures 2a,b with a brief explanation in line 110.

Indeed, there are visible differences between AHPA-L and AHPA-D AFM topography images. It is well known [Add reference] and well documented that AHPA-L chiral molecules are easier to synthesize than AHPA-D and therefore the purity of AHPA-D is lower. This difference results in reduced quality of the self-assembled monolayer of the AHPA-D molecules as seen in the AFM topographical image.

To clarify this issue a comment regarding the difference in the AFM images between AHPA-L and AHPA-D was added in lines 78-80. The referee is also right that we did not indicate directly that the AFM and corresponding MFM images display the exact same probing area. This was added in yellow marking in lines 77-78, 112-113, 431-432, 442-443.

c) 2) How were the Hall measurements performed? The manuscript shows data of change of 'Hall resistance' upon drop-casting of AHPA-L and AHPA-D solutions as a function of TIME. Is the data in Fig 3c simply a change of V_y upon molecule deposition without an applied magnetic field? It is well known that even in lithographically patterned Hall bar geometry, V_y always contains a longitudinal component in addition to the transverse Hall component. To obtain the Hall component, one needs to extract the anti-symmetric component of the V_y vs B curve.

Without a complete R_{xy} vs B curve, one cannot definitively attribute the ‘change of Hall resistance’ to variation of the anomalous Hall component; this change could originate from variation of the longitudinal component or the ordinary Hall effect due to a change in the carrier density upon molecular adsorption."

Author's response:

We thank the referee for this comment and we feel that indeed the Hall measurements procedure was not clear. It is important to note that the molecules were not drop casted during the time presented in Figure 3c. Initially device was calibrated without the molecules (differential Hall measurement method further explained in ref. 15: *Ben Dor, O., Morali, N., Yochelis, S., Baczewski, L. T. & Paltiel, Y. Local light-induced magnetization using nanodots and chiral molecules. Nano Lett. 14, 6042–6049 (2014)*), only then molecules were drop casted.

Only after molecules were chemisorbed onto the gold layer and the magnetization induced by molecular proximity effect was in equilibrium, i.e. inducing a constant magnetic field on top of the Hall channel, measurements were then made comparing the Hall voltage on the same device with and without the molecules.

The final result, as shown in Figure 3c, is the subtraction of the initial molecules-free state from the state with the molecules. The time scale in Figure 3c shows that the molecular induced magnetic field remained constant and stable throughout the measurements. A clarification regarding the Hall measurement method was added in lines 98-105, 133-135.

- d) 3) *Similarly, without showing any $M(H)$ data, the manuscript claims changes of the magnetization by approximately +35% for AHPA-L and AHPA-D adsorption respectively. Moreover, the authors attributed the magnetization changes to the rotation of the magnetic easy axis due to the chiral molecule adsorption. The $M(H)$ data need to be presented and analyzed in the manuscript in order to back up these claims. Also, the numbers quoted for ‘magnetization’ (e.g., line 143) are in fact for ‘magnetic moment’. It would be more informative to present the actual magnetization. Especially, what is the induced moment change per molecule?"*

Author's response:

We thank the referee for indicating the missing $M(H)$ data. This data was added as inset of Figure 3c and lines 136-139. Also, we thank the referee for the helpful remark regarding the numbers quoted for magnetization. The term 'magnetization' was changed to 'magnetic moment'. The information regarding the induced magnetic moment per molecule was added in lines 176-180.

e) *In summary, in my judgment, the central result reported in the manuscript is very interesting and potentially important, however, it needs to be established with more rigor. Some of the key experimental data should be presented in the manuscript and discussed in more depth. As is, I do not recommend publication of the manuscript."*

Author's response:

We wish to thank the referee once more for his/her report. We have added the corrections added to this manuscript according to the remarks given.

Reviewer #2

a) *The paper reports that magnetization switching can occur when chiral molecules are adsorbed on a ferromagnetic surface. This effect is quite interesting and follows the effects already reported by Naaman and co-workers about the so called chiral-induced-spin selectivity. I recommend acceptance of the paper."*

Author's response:

We thank the referee for a very positive review.

b) *Still, I have some questions that should be clarified before an eventual publication of these results:*

1) *The authors check this magnetization reversal using various techniques (MFM, anomalous Hall effect, SQUID-VSM), but in each case the FM structure is different and also the nature of the ferromagnetic layer (Co or Ni). The authors should explain the reasons of these choices."*

Author's response:

We thank the referee for this important comment. There are two reasons for this variety in the utilized FM materials: 1) We wanted to show that the effect is general and similar for many different types FM layers. 2) Technical issues prevent the use of all samples in all three types of magnetic measurements. The Co based thin samples were grown by MBE method and as such are more difficult to pattern without damaging the sample. Therefore they were not used in the AHE measurements. The Ni used in the AHE samples was evaporated by conventional e-beam deposition and consequently they were much easier to pattern and measure in that configuration. A remark discussing these reasons was added in lines 90-94.

- c) 2) In all cases the layer in direct contact with the SAM molecules is a gold layer. Why is this layer needed? What happens if this layer is removed? As it is now, it seems that the combination gold-chiral molecules is the required ingredient to observe such an effect."*

Author's response:

The referee raises an important issue. The gold used throughout the manuscript as an interfacial layer, acts as a protective layer against FM sub-layer oxidation. Furthermore, the top gold layer supports an ordered monolayer structure and allow for the covalent bonding between the molecules thiol end and the gold surface which is essential for the SAM formation. When the gold is removed we see a smaller effect and samples appear less stable, but this could be again attributed to oxidation. An explanation was added in lines 58-59, 328-329.

Reviewer #3

- a) Manuscript describes experimental manifestation of a new effect of magnetism induced by proximity of adsorbed chiral molecules (MIPAC). Authors claim that charge transfer between adsorbed molecules and ferromagnetic substrate together with spin selectivity induced by molecules' chirality are the origins of the demonstrated magnetization switching in the substrate.*

The finding is very exciting and clearly new: the observed effect does not require either magnetic fields or spin polarized currents to induce the switching. Moreover, authors estimate that transfer of only 10^{13} electrons per cm^2 is required to induce switching contrary to spin current densities of 10^{25} electrons per cm^2 s in the usual switching setup. In my opinion, the work clearly deserves to be published in the Nature Communications."

Author's response:

We wish to thank the referee for the warm and encouraging words.

b) There is a minor issue the authors may want to clarify before the publication. There are two statements in the manuscript. First (on p.4) is about finite lifetime of the induced magnetization (for samples not fully covered with the chiral molecules it diminishes in nine days)."

Author's response:

The referee raises an important issue which we yet not fully understand. We think that because there exists a coupling between the neighboring FM layer and the chiral monolayer, the induced magnetization reorientation stems from a molecular tilted state at an angle to the original easy axis, when samples are partially covered. In what follows, this "magnetization decay" is attributed to the exchange interaction that causes the domains outside the selective adsorption areas to gradually reorient the molecular induced magnetization back to an average state due to a weaker effect derived from molecular tilting. Our suggested explanation is presented in lines 213-220 in the manuscript.

c) Second (on p.5) is about inducing magnetization in small areas (down to a single domain size) with a prospect of utilizing this local magnetization to build surface memory logic. It is a bit confusing if the magnetization induced locally will be stable. And if yes, what is the difference from the situation mentioned in the first statement? Is it due to domain formation?"

Author's response:

We thank the referee for this important comment. We agree with the referee that it might be confusing to suggest localized surface memory whilst having gradual decay in anisotropy. The answer here lies in the samples at hand. Further investigation is required in other samples and different linkers that may yield different interactions and consequently longer decay mechanisms. We added this explanation in the summary in lines 219-220.

We thank all 3 referees and the editor for their very positive reports, and for their useful and important comments. We were happy to notice that all 3 referees are in favor of publishing the paper after minor corrections.

It what follows we respond point-by-point to the referees' comments.

REVIEWERS' COMMENTS:

Reviewer #1 (Remarks to the Author):

I am satisfied with the authors' technical responses to my review. I recommend publication of the manuscript. There are still many grammatical errors and other language issues in the manuscript, I suggest a careful editing before publication.

Author's response:

We thank the referee for supporting our paper for publication. The final revised manuscript overwent professional linguistic editing and is therefore highly suitable for publication.

Reviewer #3 (Remarks to the Author):

Authors answered all the questions. The manuscript can be recommended for publication.

Author's response:

We thank the referee for endorsing our paper for publication.